# Performance Assessment of the Post-Tensioned Anchorage Zone Using High-Strength Concrete Considering Confinement Effect

**DOI:** 10.3390/ma14071748

**Published:** 2021-04-02

**Authors:** Jun Suk Lee, Byeong Hun Woo, Jae-Suk Ryou, Jee-Sang Kim

**Affiliations:** 1Concrete Laboratory, Department of Civil and Environmental Engineering, Hanyang University, 222, Wangsimni-ro, Seongdong-gu, Seoul 05510, Korea; free_gopa@naver.com (J.S.L.); dimon123@hanyang.ac.kr (B.H.W.); jsryou@hanyang.ac.kr (J.-S.R.); 2Concrete Laboratory, Department of Civil and Architectural Engineering, Seokyeong University, 124, Seogyeong-ro, Seongbuk-gu, Seoul 02713, Korea

**Keywords:** post-tension, anchorage zone, reinforcement, high-strength concrete, confinement effect

## Abstract

Post-tensioned anchorage zones need enough strength to resist large forces from jacking forces from prestress and need spiral reinforcement to give confinement effect. High-strength concrete (HSC) has high-strength and brings the advantage of reducing material using and simplifying reinforcing. We tested strain stabilization, load–displacement, and strain of lateral reinforcements. Specimens that used one and two lateral reinforcements without spiral reinforcement did not satisfy the strain stabilization. Load capacity also did not satisfy the condition of 1.1 times the nominal tensile strength of PS strands presented in ETAG 013. On the other hand, specimens that used three and four lateral reinforcements without spiral reinforcement satisfied the strain stabilization but did not satisfy 1.1 times the nominal tensile strength of PS strands. However, the secondary confinement effect could be confirmed from strain stabilization. In addition, the affection of HSC characteristics could be confirmed from a reinforcing level comparing other studies. The main confinement effect could be confirmed from the reinforcement strain results; there was a considerable difference between with and without spiral reinforcement at least 393 MPa. Comprehensively, main and secondary confinement effects are essential in post-tensioned anchorage zones. In addition, the performance of the anchorage zone could be increased by using HSC that the combination of high-strength and confinement effect.

## 1. Introduction

There is a recent trend of reducing the materials and increasing concrete performance [1]. High-strength concrete (HSC) is a suitable material for this trend. In HSC common cases, it has a low w/c ratio and brings a high-strength for cement composites. The cement matrix is denser than normal concrete (NC) [2]. HSC uses materials less than NC, but HSC shows higher strength and durability than NC [2,3,4]. HSC has a good carbonation resistance due to having a dense cement matrix and high compressive strength [5]. In addition, HSC shows good resistance to chloride penetration and freezing–thawing [3,6]. Not only does it have strength and durability aspects, but HSC is also used widely in the construction of high-rise buildings [7] and long-span bridges [8]. In this way, HSC is an essential material for the construction field.

Girders especially use the post-tension method when construct bridges, the section of the anchorage zone goes to be thick, as shown in Figure 1. This means that great amounts of materials, such as cement and lateral reinforcements, are used in the anchorage zone to endure large jacking forces caused by post-tension. However, section size can be reduced by using HSC. Davari et al. [9] confirmed columns’ performance in earthquake conditions, which used 50 MPa and 100 MPa concrete. The column sections were 500 × 500 mm^2^ of 50 MPa concrete and 400 × 400 mm^2^ of 100 MPa concrete. The 100 MPa of concrete column showed far better performance in earthquake conditions. Davari et al. [9] found that using HSC brings economic effect through reducing reinforcements. Hussain et al. [10] studied the possibility of reducing pavement thickness using NC and HSC. Pavement thickness is a kind of concrete member size. Hussain et al. [10] confirmed that pavement thickness could be reduced by using HSC. Therefore, HSC has many benefits in the construction field; in other words, reinforcements can be reduced by using HSC at the same section size.

Typically, a post-tensioned anchorage zone requires many reinforcements to endure the tensile force of bursting strain. Using many reinforcements makes sections of the concrete members thick and brings rising costs and inefficient materials use. Ro et al. [11] designed a beam using the post-tension method and found that lateral reinforcement was intensively arranged in the location of the bursting strain zone. In addition, specimens that used lateral reinforcements in the bursting strain zone showed low bursting stress when the jacking force was introduced. Mao et al. [12] performed an experiment of the post-tension method on the slab of a box girder. The introduced jacking forces were very large: 6054 KN at the top slab and 3711 KN at the bottom slab. In addition, Mao et al. [12] used a different number of anchorage zones on a slab. Cracks on a concrete surface were found even when the compressive strength of concrete was 55 MPa. Reinforcing was enough; however, jacking forces were quite large, and torsion was generated because of different anchorage numbers. These are good methods to evaluate the performance of post-tension anchorage zones through large-scale experiments. However, there is a simple method for evaluating the performance of post-tension anchorage zone: the ETAG 013 method [13]. According to the ETAG 013 guidelines, the design of a test specimen is simple. Kim et al. [14] performed the ETAG method for evaluating post-tension performance using newly designed anchorage equipment, and they showed well how to the ETAG 013 experiment conduct. Kim et al. [14] showed the shape effect of anchorage plate through load transfer test. The compressive strength of concrete was 30 MPa, and all specimens showed large strain behavior in loading conditions. This was because they did not use spiral reinforcement. Usually, the post-tension method uses spiral reinforcement to give an effect of confinement to resist large jacking forces from post-tension. Theoretically, the confinement effect in concrete brings a 1.5 to 2 times increase of compressive strength. Huang et al. [15] showed the confinement effect through a simple experiment. With the confinement effect, the load-bearing tolerance increases and this effect can be found easily. Marchão et al. [16] clearly showed the confinement effect in load transfer tests of the ETAG method. Marchão et al. [16] compared the performance of NC and high-performance fiber-reinforced concrete (HPFRC). From the results, HPFRC showed a much larger load-bearing tolerance than NC. The authors analyzed the confinement from the spiral and lateral reinforcements in detail. Spiral reinforcement gives the main confinement effect; this means that spiral reinforcement gives a much larger confining performance than lateral reinforcements. This is easily confirmed from the study of columns; the confinement effect by the spiral reinforcement is very important [17]. However, secondary (small confinement effect from lateral reinforcements) confining cannot be ignored that the secondary confining also helps to increase the bearing tolerance [18]. The theoretical details are presented in Section 2. With confinement effects from the spiral and lateral reinforcements, high-strength (upper than 2160 MPa) prestressing (PS) tendons can be applied to post-tensioning members. According to Yang et al. [19], the applicability of high-strength PS tendons (2360 MPa) could be used in concrete members with confinement effect even the compressive strength of concrete was 27.6 MPa.

Many studies have performed good experiments and analyses of post-tensioned concrete members. However, there are no studies considering the amount of change of reinforcements on performance. The amount of lateral reinforcements is important in that they give a secondary confinement effect. Hence, changing the number of lateral reinforcements must be studied because the performance should be changed with the number of lateral reinforcements. In addition, the possibility of reducing reinforcements has already been examined in previous studies [9,11,12]. A study for using reinforcements effectively with increasing concrete strength should be studied to follow the recent trend of material used. In this study, the load transfer performance of post-tension anchorage members using HSC was evaluated by changing the number of lateral reinforcements and with/without spiral reinforcement. Usually, the previous studies checked the effect of the shape of the bearing plate or confirmed the performance of the developed anchorage system [11,14,20]. Therefore, it was considered that the confinement effect with HSC and simple reinforcing should be determined. From the setting of the experiment, the confinement effect and the advantage of using HSC were evaluated. To determine the purpose of this study, the strain of the bursting site and vertical displacement were measured for assessing the performance of specimens. The load transfer method followed ETAG 013 method [13].

## 2. Materials and Experimental Program

### 2.1. Materials

Type-I ordinary Portland cement (OPC) was used in this experiment [21]. In addition, blast furnace slag (BFS) and silica fume (SF) were used to give a filler effect for increasing the strength of concrete [22,23]. The 100 MPa of concrete was used in this experiment. The mix properties are indicated in Table 1.

FA and CA are fine and coarse aggregates, respectively. SP is a superplasticizer; AE is an air-entraining agent. SP used 3% of cement weight, and AE used 0.06% of cement weight. The properties of FA, CA, SF, and BFS are summarized in Table 2.

Because of the low W/B ratio, using SP was necessary to secure sufficient workability. A slump flow test was performed to check the workability of concrete; the result was 680 mm. According to the result of slump flow, the workability was suitable. Specimens were demolded after 1 day of air curing and performed 60 °C of steam curing for 3 days. After steam curing, strength was evaluated. The results of strength are summarized in Table 3.

For designing specimens, D10 and D16 reinforcements were used that had a nominal diameter of 9.53 mm and 15.7 mm, respectively. The yield strength (f_y_) of reinforcements was the same value of 400 MPa (yield strain = 0.002). The D10 reinforcement was used as vertical and lateral reinforcement. The D16 reinforcement was used as the spiral reinforcement. The details of reinforcements are shown in Figure 2.

For making specimens, an anchorage kit was needed to give the details in specimens. Therefore, VSL TYPE EC 5-12 (VSL Korea, Seoul, Korea) anchorage kit was used in this study [24]. The details are shown in Figure 3.

Although tendons are indicated in Figure 3, tendons were not used in this study. Except for tendons, all sets of anchorage kits were used in this experiment.

### 2.2. Specimen Design

We followed the design guide of the ETAG 013 [13] to design specimens. According to the ETAG design guide, many parameters were limited for designing specimens, such as specimens’ dimensions and using the number of reinforcements. The conditions of section dimension of concrete are indicated in Equations (1) and (2) [13]:A_c_ = x ∙ y = a ∙ b(1)
x ≥ 0.85a and y ≤ 1.15b(2)
where, A_c_ is a section of the specimen (mm^2^), a and b are the side lengths of the specimen (mm), x and y are minimum specified center spacing of the particular PS strands in the structure or specified edge distance of PS strands (mm). Dimensions of specimens should meet the limit of Equations (1) and (2). The height of the specimen was more than 2 times the longer length of the side of the section.

The limit conditions of reinforcement placing were as follows:The section area of vertical reinforcement was less than 0.003 A_c_;The number of stirrups was less than 50 kg (steel kg/concrete m^3^).

According to the condition of specimen designing, specimens were designed like Table 4 and Figure 4. Members were a total of eight specimens in two cases, which are with/without spiral reinforcements. Figure 5 explains the naming of specimens.

The bottom reinforcing in Figure 5 was necessary because a large load would be applied to the specimens, leading to large compressive stress on the bottom side. To evaluate the performance of specimens precisely, the bottom side of specimens must not be demolished. Hence, bottom reinforcements were installed.

All of the reinforcements give the confinement effect that the spiral reinforcement gives the main confinement. The lateral reinforcements give a secondary confinement effect. The location of introducing the jacking force of post-tensioning burdens very large compressive stress. Therefore, a confinement effect is necessary, and spiral reinforcement can give main confining. It brings 1.5 to 2 times of increasing of compressive force resistance [15]. Although secondary confining by lateral reinforcements is relatively small than the spiral reinforcement, it increases the jacking forces’ load resistance [16]. The mechanism of confining is indicated in Figure 6. The confinement effect is hard to calculate, but it can be confirmed by checking the strain of lateral reinforcements in the loading condition [16].

### 2.3. Loading and Sensing Plan

In this experiment, 3000 KN of universal test machine (UTM) was used. The loading plan was followed by the ETAG 013 of the load transfer test. The load transfer test of Figure 7 is a kind of repeating loading test. This process is for strain stabilizing of specimens [13]. F_pk_ in Figure 7 is the ultimate strength of PS strands, whose fully applied amount in the anchorage zone and F_pk_ in this experiment was 2073.53 KN. In this paper, PS strands were not used. Therefore, the load value of F_pk_ was assumed as same as using twelve PS strands of SWPC-7B 12.7 mm class. The details of the assumed PS strand are indicated in Table 5. From the assumed condition, the load value was calculated (Table 6).

When loading, the load was not fully applied for machine safety reasons. Although the capacity of UTM is 3000 KN, the load was applied to 2550 KN (1.23 F_pk_), or 86% of UTM capacity. According to Figure 7, the load transfer test process can be explained. First, we applied a load up to 20% of the F_pk_. Second, we increased the load gradually that 40%, 60% and 80% of the F_pk_. After repeating this at least 10 times, the loading process was increased up to 12% and 80% of F_pk,_ and this experiment was repeated 10 times. After repeating the load, we increased the load until the specimen failed. In this experiment, the load condition, which was set as 1.23 F_pk,_ could not reach the specimen’s failure. Therefore, the failure condition was assumed due to the lateral reinforcements caused by strain measurement. The basis of assuming the failure condition was the design codes of concrete [25,26]. Design codes of concrete structures assumed the failure state that reinforcements were caused by external loads. According to these codes, the failure condition was set as the point of reinforcement yielding.

To evaluate specimens’ performance, strain and displacement sensors were used. The sensing plan is shown in Figure 8. Reinforcement strain sensors were installed on two sides of lateral reinforcements and spiral reinforcement. In addition, the bursting strain was measured by the installation of surface strain sensors. The most important factors are displacement and load. Therefore, LVTDs were installed on two sides of the specimen. The schematic process of this study is summarized in Figure 9.

As shown in Figure 8, there are two conditions for checking the strain stabilization work load test. In the ETAG 013, strain and crack are used to check whether to continue the load transfer test or not [13]:W_n_ − W_n−4_ ≤ 1/3(W_n−4_ − W_0_), n ≥ 10(3)
ε_n_ − ε_n−4_ ≤ 1/3(ε_n−4_ − ε_0_), n ≥ 10(4)
where W is crack width (mm), ε is measured strain. In addition, design codes recommend that the post-tensioned anchorage zones do not generate cracks [25,26]. Nevertheless, cracks were generated, and crack width was checked to determine if the crack width exceeded 0.2 mm [25,26], and crack had not been generated at the first stage. If the specimen did not meet the strain stabilizing condition of Equations (3) and (4) and crack width, the load was introduced continuously until yielding the lateral reinforcements without strain stabilization for comparing other specimens. To evaluate the performance of specimens, a total of 5 factors were checked:Factor 1. Did cracks that occurred at the first stage satisfy the strain stabilization?Factor 2. Does a crack-width exceeding 0.2 mm during strain stabilization work?Factor 3. Load and displacement;Factor 4. Bursting strain;Factor 5. Reinforcement strain.

These factors were measured in this study and evaluated the performance of specimens. In addition, the crack width was averaged when evaluating strain stabilization work. The strain stabilization work was repeated 10 times.

## 3. Result and Discussion

### 3.1. Strain Stabilization Result

SP-N-LT-1 and SP-N-LT-2 specimens did not satisfy factor 1. Specimens should not generate cracks during the first loading stage. However, according to Figure 10a, a crack was detected at the first loading stage in the SP-N-LT-1 and SP-N-LT-2 specimens. Therefore, loading work continued following the condition of this paper and SP-N-LT-1 and SP-N-LT-2 specimens showed the performance that the specimens could not exceed 1.1 F_pk_ of the least recommendation performance of the ETAG 013 [13]. Hence, SP-N-LT-1 and SP-N-LT-2 showed not enough performance of post-tensioning. On the other hand, SP-N-LT-3 and SP-N-LT-4 satisfied factors 1 and 2 conditions of this experiment. The SP-N-LT-3 specimen showed a crack point at the third stage during loading work. According to Equation (3), the crack width had to be checked at the 10th stage, 6th stage, and W_0_ is 0. W_10_ at 0.8F_pk_ was 0.124 mm, and W_6_ at 0.8F_pk_ was 0.106. The calculation showed that W_10_ − W_6_ was 0.018 mm and 1/3(W_6_ − W_0_) was 0.0353 mm. In addition, the SP-N-LT-4 specimen showed the cracking point after strain stabilization work. Therefore, two specimens, which were SP-N-LT-3 and SP-N-LT-4, performed the strain stabilization work. SP-N-LT-3 specimen reached almost 1.1 F_pk_ after strain stabilization. This result was the best performance among the SP-N series. SP-N-LT-4 specimen showed the cracking point after strain stabilization work and almost reached 1.03F_pk_. It was expected that the SP-N-LT-4 would show the highest load-resistant performance, but this specimen barely exceeded 1.0 F_pk_. However, cracking performance was the best among the SP-N series. From the results of the SP-N series, the secondary confinement effect could be confirmed from the experiment. The SP-N-LT-1 and SP-N-LT-2 specimens could not show enough performance. However, SP-N-LT-3 and SP-N-LT-4 specimens showed satisfying factor 1 and 2 conditions, and it is considered that the effect of secondary confinement effect. This phenomenon supports the study of Kim et al. [14]. In the study of Kim et al. [14], a 1.1 F_pk_ was set at 284 KN, and load test results exceeded 1.1F_pk_. These specimens did not use spiral reinforcement but used lateral reinforcements.

For evaluating the strain stabilization, strain data are also important. The bursting strain was used in this study. Kim et al. [14] evaluated strain stabilization using strain values well. Therefore, we followed this calculation with Equation (4). SP-N-LT-1 and SP-U-LT-2 specimens did not satisfy factor 1 and 2 conditions. Therefore, evaluating the strain stabilization with bursting strain was not included, and the results of bursting strain measurement were indicated in Figure 11. In the case of SP-N-LT-3, strain values were derived that ε_0_, ε_6_, and ε_10_ were 0.000232, 0.000257, and 0.00026, respectively. Following Equation (4), the calculation showed that ε_10_ − ε_6_ was 3 × 10^−6^ and 1/3 (ε_6_ − ε_0_) was 8.33 × 10^−6^. In addition, SP-N-LT-4 showed the busting strain that ε_0_, ε_6_, and ε_10_ were 0.000151, 0.000248, and 0.000275, respectively. The calculation showed that ε_10_ − ε_6_ was 2.7 × 10^−5^ and 1/3 (ε_6_ − ε_0_) was 3.23 × 10^−5^. Therefore, SP-N-LT-3 and SP-U-LT-4 specimens met the condition of Equation (4) and strain stabilization. In the same sense, the SP-U series also satisfied the condition of Equation (4) and strain stabilization. It can be confirmed that the bursting strain decreases gradually according to an increase in the lateral reinforcements. This trend can be considered that the combination of confinement effect from lateral and spiral reinforcements. Another study clearly showed the same trend of bursting strain in this study. The specimens of the study of Kim et al. [27] showed that bursting strain decreased when specimens were reinforced by the spiral reinforcement [27]. In addition, the location of maximum bursting strain moved, and the specimens of this study also moved. The details are indicated in Figure 12. Theoretically, maximum bursting strain is generated at the location of 10–30% of member height from the top surface of the member. This was studied by Robinson et al. [28]. The maximum bursting strain of the SP-N-LT-1 was generated at the location of 0.16H, and the other series of the SP-N was generated at the location of 0.19 H. In addition, the maximum bursting strain location of the SP-U-LT-1 and SP-U-LT-2 specimens appeared 0.19 H and SP-U-LT-3, and SP-U-LT-4 specimens were 0.22 H. The location was going downward as reinforcing increased. It can be considered that the location of maximum bursting strain was going down as a function of the increasing confinement effect.

This trend was also shown in the study of Kim et al. [27]. In this, the locations of maximum bursting strain were 0.3 H in the case of non-reinforced specimens. However, the locations of the maximum bursting strain of reinforced specimens using spiral reinforcement were 0.5 H [27]. Therefore, the trend of this study supports our results well [27,28].

### 3.2. Load–Displacement and Reinforcement Strain Results

The load was introduced continuously without repeating the work on the SP-N-LT-1 and SP-U-LT-2 specimens because these specimens did not satisfy the strain-stabilization condition. However, other specimens did repeat loading that 0.12 F_pk_ to 0.8 F_pk_. Load–displacement results are shown in Figure 13.

There was a loading restriction because of the machine safety; this experiment had a clear limitation of observing the max capacity of each specimen. SP-N-LT-1 and SP-U-LT-2 specimens failed almost at the point of 0.8 F_pk_. This means that the least reinforcing level is needed. The jacking force introduces approximately 0.7 f_pu_ or 0.8–0.9 f_py_ [26]. This value is usually smaller than 1.0 F_pk_. Therefore, specimens must resist the load to at least 1.0 F_pk_. Hence, the least-reinforcing-level of specimens without spiral reinforcement is as same as the SP-N-LT-3 using 100 MPa of HSC. In this part, the advantage of using HSC clearly appeared. Compared to other studies, the reinforcing level of this study was lower [11,12,14,16,19,20]. In particular, Marchão et al. [16] installed lateral reinforcements with 60 mm spacing, and the size of specimens was similar to this study. Yang et al. [19] installed lateral reinforcements with 65 mm spacing and specimen sizes almost 2 times more greater than this study. In this aspect, HSC brings the effect of reducing the use of reinforcements. This was demonstrated by the results of the SP-N-LT-3 and SP-N-LT-4 specimens. In addition, a post-tensioned anchorage zone using HSC can bring enough performance with the spiral reinforcement, even reducing lateral reinforcements. All cases of the SP-U series met the 1.23 F_pk_ goal load of this experiment. Considering the gradient after strain stabilization work, specimens have the margins for resisting the load upper than 1.23 F_pk_. Comparing the results of Marchão et al. [16] and Kim et al. [14,27], specimens can resist at least 1.6 F_pk_. In addition, it was demonstrated that the main confinement effect by the spiral reinforcement was very important to increase the performance of anchorage zones from the experimental results of the SP-U series.

Failure condition was set as the yielding point of lateral reinforcement in this study. The yield strength of lateral reinforcements was 400 MPa, and the strain value was 0.002. During loading conditions, all strain values of lateral reinforcements were checked. The results are presented in Figure 14.

All of the SP-N series showed yielding of lateral reinforcements. The location is indicated in Figure 14. The location of yielded lateral reinforcement was the maximum bursting strain. This location is indicated in Figure 12. However, the load capacity was increased at the point of yielding strain. This was also related to the confinement effect and characteristics of HSC. This was the same meaning with load–displacement results. First, SP-N-LT-1 and SP-N-LT-2 specimens had a lack of confining by lateral reinforcement due to the smaller reinforcing than LT-3 and 4 specimens. However, SP-N-LT-3 and 4 showed satisfying strain stabilization and exceeding 1.0 Fpk. These results mean that the SP-N-LT-3 and SP-N-LT-4 specimens had the poorest performance, even smaller reinforcing and without spiral reinforcement, comparing other studies [11,12,14,16,19,20,27]. In the same way, the combination of HSC and confinement effect by lateral reinforcement (secondary confining) can be confirmed from the results of Figure 14b. HSC makes members that could apply the less reinforcing; lateral reinforcement gives a secondary confinement effect. Therefore, this combination brought a decreasing yielding strain value.

From the results of the SP-U series, the main confinement effect of the spiral reinforcement could be found. Comparing to the SP-N series, strain values were much smaller than the SP-N results. All specimens of the SP-U did not show the reinforcements yielding. The elastic modulus of reinforcements was 202 Gpa. The results could be converted in stress in 20.6 MPa of the SP-U-LT-1, 13.4 MPa of the SP-U-LT-2, 19.4 MPa of the SP-U-LT-3, and 7.64 MPa of the SP-U-LT-4. These stress values were the maximum stress of each specimen. On the other hand, reinforcement stress of the SP-N series appeared that 478 MPa of the SP-N-LT-1, 552 MPa of the SP-N-LT-2, 434 MPa of the SP-N-LT-3, and 414 MPa of the SP-N-LT-4. Therefore, it can be calculated that the stress compensation was large, approximately at least 393 MPa. Therefore, it was demonstrated that the main confinement effect by the spiral reinforcement was very important for the performance of post-tensioned anchorage zones.

## 4. Conclusions

This study focused on assessing the performance of the post-tensioned anchorage zone considering the effect of confinement effect with a 100 MPa class of HSC. Many effects were confirmed from this experiment. From the results, the possibility of reducing the number of lateral reinforcements was confirmed. In addition, the importance of the secondary confinement effect was also confirmed. According to the results of this study, comprehensive conclusions are as follows:Strain stabilization work followed Equations (3) and (4). SP-N-LT-1 and two specimens did not satisfy the condition of strain stabilization. However, SP-N-LT-3 and SP-N-LT-4 satisfied the condition of strain stabilization. We considered this to be due to the combination of high-strength characteristics of HSC and secondary confinement effect;The importance of the main confinement effect by the spiral reinforcement was confirmed by experiment results. Bursting strain results of the SP-U series were smaller than the SP-N series, and the maximum location of bursting strain moved to the downside comparing to the SP-N series. This behavior was considered the affection of the main confinement effect. In addition, the main confinement effect was able to confirm clearly in the results of lateral reinforcements strain. Lateral reinforcements of the SP-N series were yielded by loading work. However, the SP-U series did not show yielding of lateral reinforcements. The stress value was much smaller than the SP-N series;Comprehensively, HSC brought the effect that specimens could reduce the number of lateral reinforcements. In addition, the importance of the secondary confinement effect could be confirmed from the results of the SP-N-LT-3 and SP-N-LT-4 specimens. The effect of the main confinement effect by the spiral reinforcement could be confirmed from the results of load–displacement and strain of lateral reinforcements.

## Figures and Tables

**Figure 1 materials-14-01748-f001:**
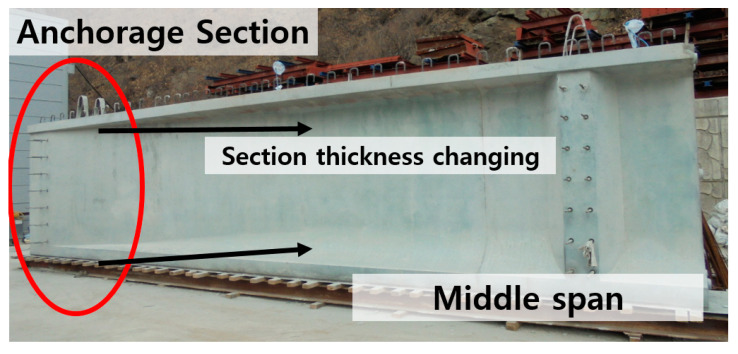
Section thickness changes between anchorage section and middle span.

**Figure 2 materials-14-01748-f002:**
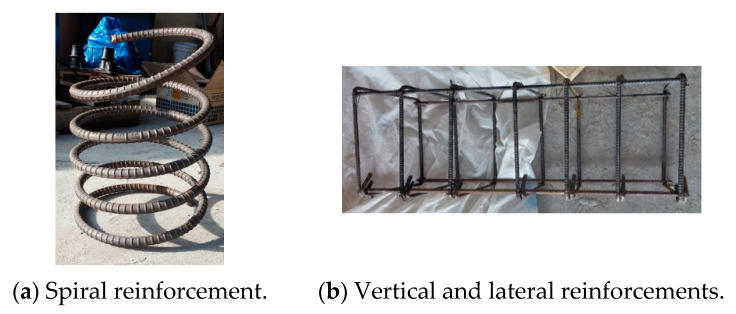
Details of reinforcements.

**Figure 3 materials-14-01748-f003:**
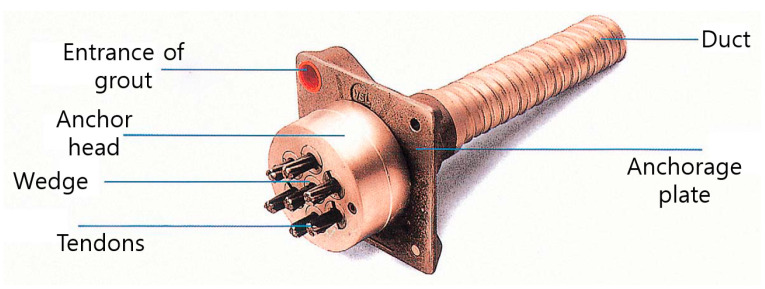
Details of anchorage set of VSL.

**Figure 4 materials-14-01748-f004:**
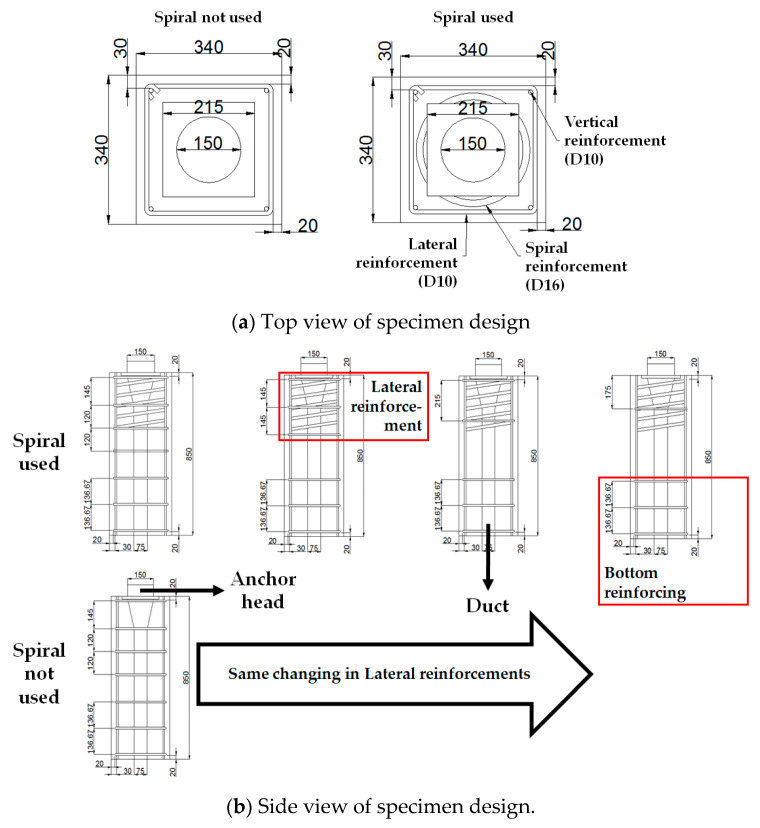
Specimen design of all cases (unit: mm).

**Figure 5 materials-14-01748-f005:**
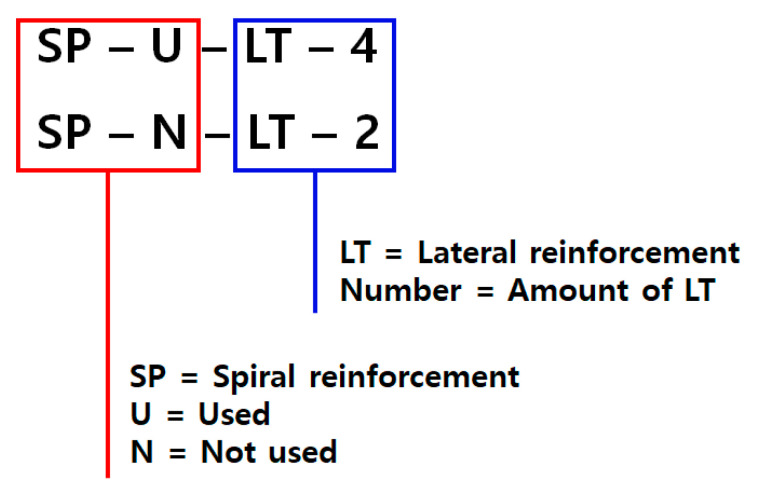
Naming of specimens.

**Figure 6 materials-14-01748-f006:**
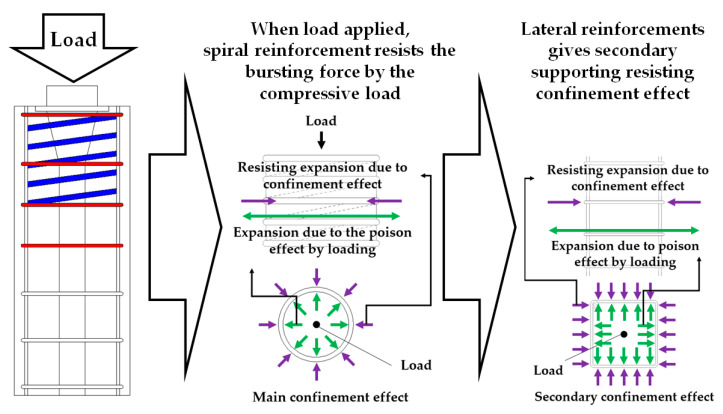
Confinement effect mechanism by the spiral and lateral reinforcements.

**Figure 7 materials-14-01748-f007:**
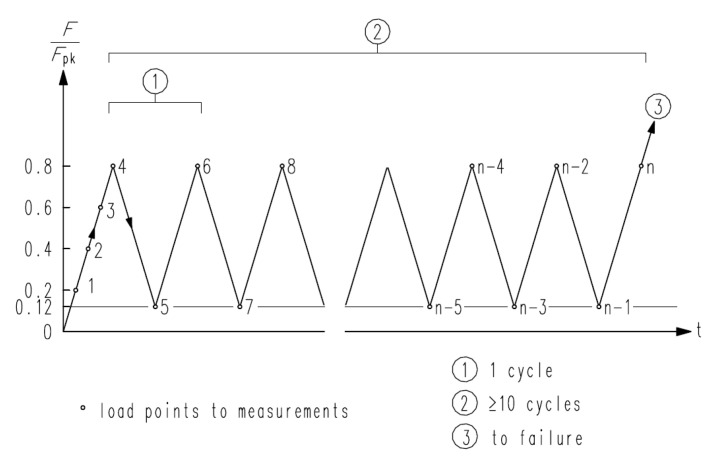
Schematic process of load transfer test [13].

**Figure 8 materials-14-01748-f008:**
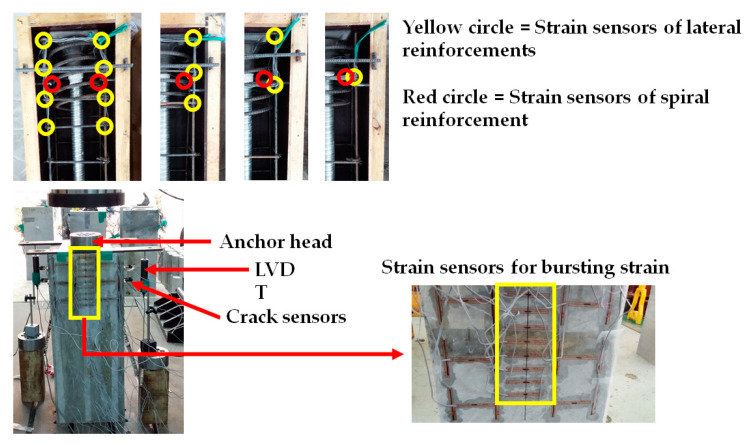
Sensing plans of the experiment.

**Figure 9 materials-14-01748-f009:**
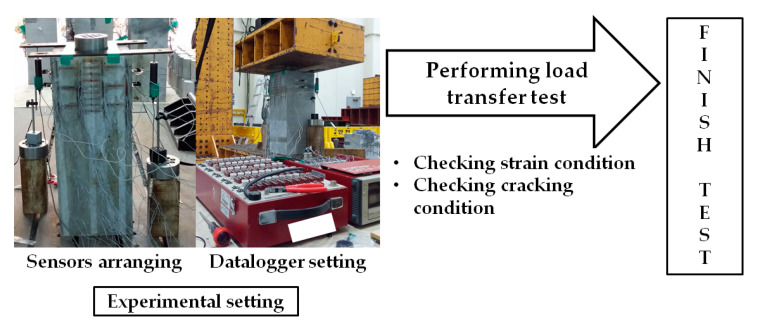
Schematic process of experiment.

**Figure 10 materials-14-01748-f010:**
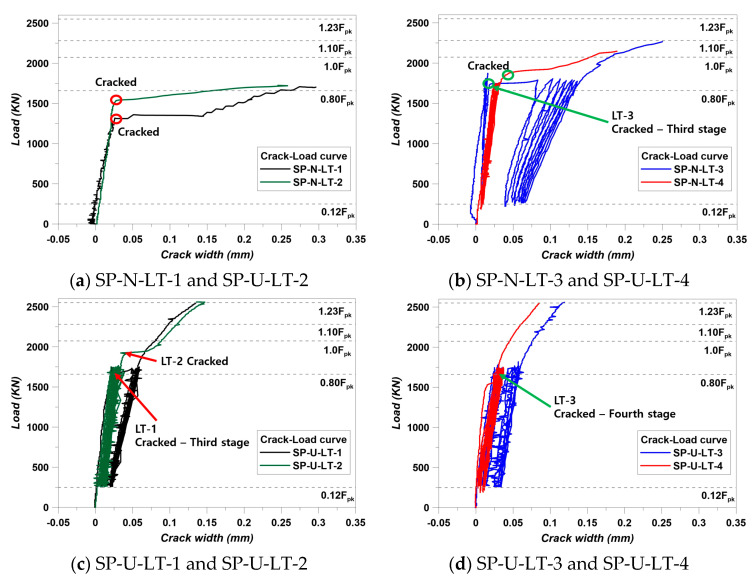
Crack–load curve of specimens: (**a**) SP-N-LT-1 and SP-U-LT-2, (**b**) SP-N-LT-3 and SP-U-LT-4, (**c**) SP-U-LT-1 and SP-U-LT-2, (**d**) SP-U-LT-3 and SP-U-LT-4.

**Figure 11 materials-14-01748-f011:**
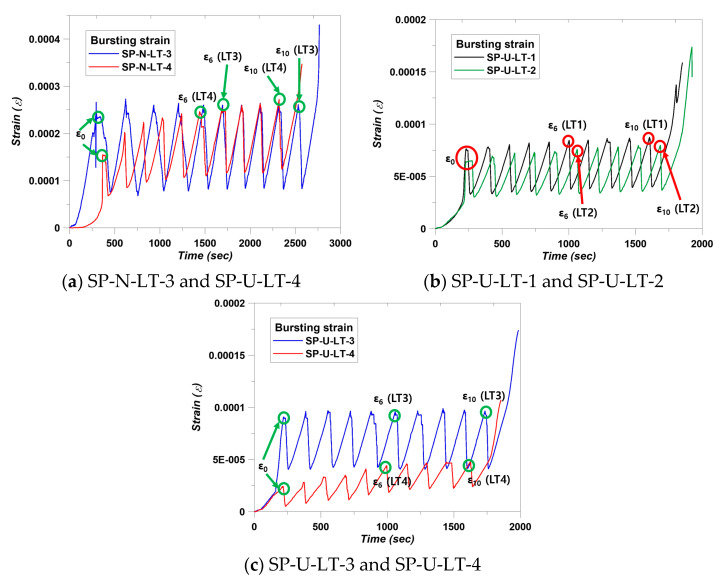
Results of bursting strain measurement.

**Figure 12 materials-14-01748-f012:**
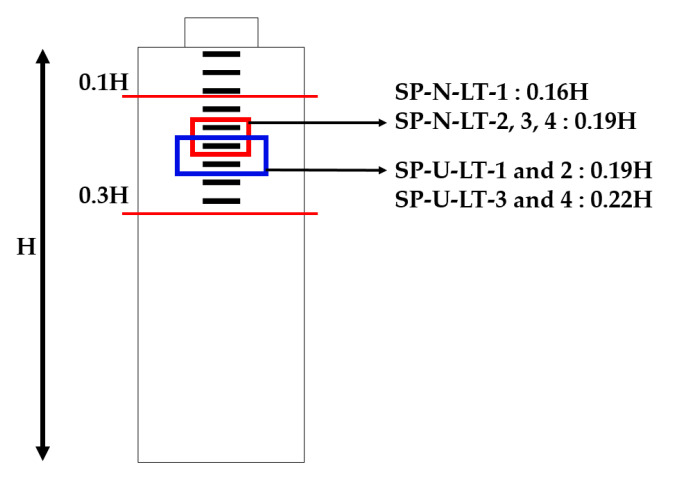
Location of the maximum bursting strain of specimens.

**Figure 13 materials-14-01748-f013:**
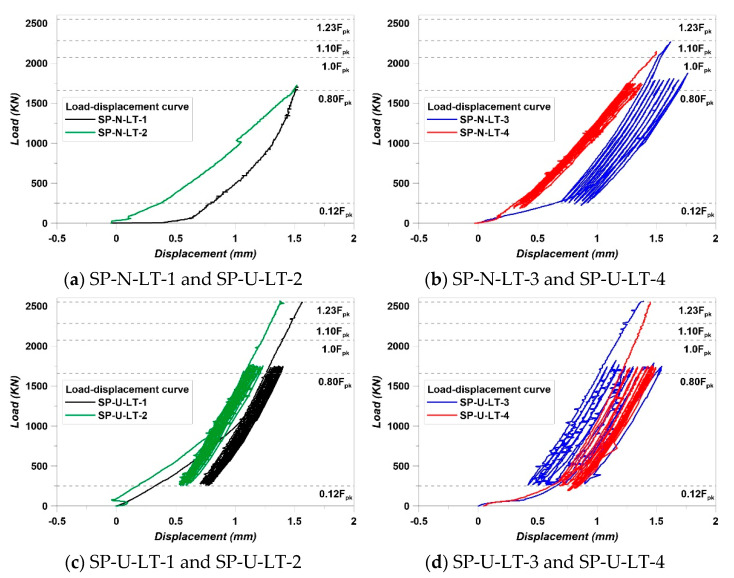
Load–displacement results.

**Figure 14 materials-14-01748-f014:**
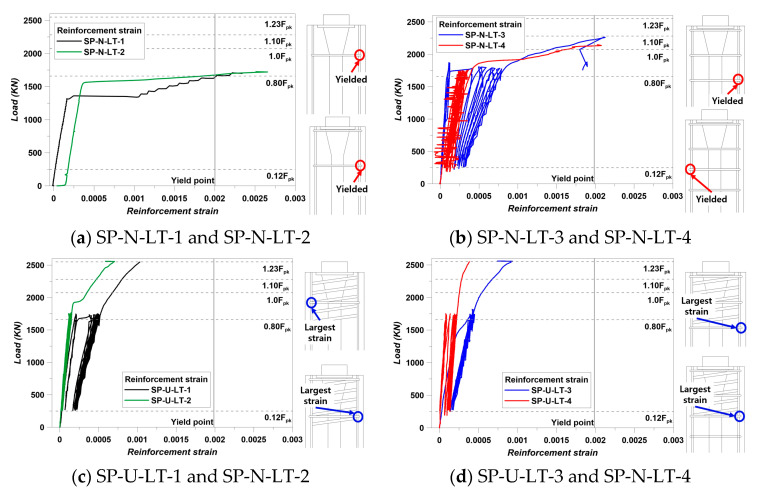
Lateral reinforcements strain results: (**a**) SP-N-LT-1 and SP-N-LT-2, (**b**) SP-N-LT-3 and SP-N-LT-4, (**c**) SP-U-LT-1 and SP-N-LT-2, (**d**) SP-U-LT-3 and SP-N-LT-4.

**Table 1 materials-14-01748-t001:** Mix properties of high-strength concrete (HSC).

W/B (%)	S/a (%)	Unit: kg/m^3^							
		Water	Cement	BFS	SF	FA	CA	SP	AE
20	42	165	578	165	680	572	792.4	18	0.35

**Table 2 materials-14-01748-t002:** Material properties.

Materials	Particle Size	Dry Density (kg/m^3^)
Fine aggregate	0.15 mm to 2.2 mm	2.62
Coarse aggregate	9.5 mm to 25 mm	2.68
Blast furnace slag	10 μm to 55 μm	2.91
Silica fume	0.1 μm to 1 μm	2.2

**Table 3 materials-14-01748-t003:** Mechanical properties of concrete.

Compressive Strength (MPa)	Split Strength (MPa)	Flexural Strength (MPa)
105.1	4.84	7.01

**Table 4 materials-14-01748-t004:** Specimen dimensions and usage of lateral reinforcements (include bottom reinforcements and except the spiral reinforcement).

Specimens	Width (mm)	Length (mm)	Height (mm)	Used Lateral Reinforcements (Unit: kg)
SP-U-LT-4	340	340	850	4.704
SP-U-LT-3				4.032
SP-U-LT-2				3.360
SP-U-LT-1				2.688
SP-N-LT-4				4.704
SP-N-LT-3				4.032
SP-N-LT-2				3.360
SP-N-LT-1				2.688

**Table 5 materials-14-01748-t005:** Details of the prestressing (PS) strand.

Assumed Strand	Nominal Section Area (mm^2^)	Nominal Tension Strength (MPa)
SWPC-7B 12.7 mm	92.90	1860

**Table 6 materials-14-01748-t006:** Applied load conditions.

F_pk_ (KN)	0.8F_pk_ (KN)	1.1F_pk_ (KN)
2073.53	1658.82	2280.9

## Data Availability

Data available on request to corresponding author.

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
