# Peer review of "Performance Assessment of the Post-Tensioned Anchorage Zone Using High-Strength Concrete Considering Confinement Effect"

_materials, 2021, doi:10.3390/ma14071748_

Round 1

Reviewer 1 Report

The manuscript is difficult to read due to the poor construction and style of English language. Extensive editing of English language and style is required.

Therefore, I will not comment on and correct any language errors, which are a lot in the manuscript.

Abstract is not informative. For research articles, abstracts should give a pertinent overview of the work. It needs to be completely rewritten. Including the names of the samples in the abstract has no value for the reader, because no one knows what they are and what their symbols mean.

I suggest that the authors present very clearly what scientific value their article brings to the public space using correct English. Without linguistic proofreading, it is impossible to understand what the main message of the Authors is.

I do not diminish the scientific / engineering value of this work, but in this form the article not only cannot be accepted, but generally correctly understood by the reviewers.

Author Response

Respected Reviewer,

Thank you for reviewing our manuscript entitled Performance assessment of the post-tensioned anchorage zone using high strength concrete considering confinement effect for possible publication in the journal of “Materials”. We are thankful to you for your quick and valuable feedbacks to improve the quality of our manuscript for possible publication in the journal. We have revised the manuscript according to your comments and suggestions. The pointwise replies are given here in the reply file.

Reviewer 2 Report

Article deal with post-tensioned anchorage zones by high strength concrete based on several tests.

The aim of the article is 100 MPa HSC and a very detailed experiment shows how this high-strength concrete can strengthen the anchor zones.

The biggest benefit can be found in the finding that HPC performs a reinforcing effect even with a smaller number of secondary reinforcements.

The scope of the article is above average and expands knowledge.

The article is very scientific and at the same time show a clear statement on all aspects of material preparation, testing and evaluation. The results of the presented research are useful and it is highly desirable that they be published.

I highly appreciate the graphic design, which helps the reader to understand everything they need.

The structure of the article is also well prepared.

I didn't find a problem.

Author Response

(The authors gave the same response as above.)

Reviewer 3 Report

This paper is original and interesting.  It is clearly and well written. 

The methodology and experiment are adequately described while the results and conclusions are clearly presented.

I suggest that the paper be accepted and published in present form. 

Author Response

(The authors gave the same response as above.)

Reviewer 4 Report

Dear Authors the subject of the paper is interesting and its purpose complies with the journal’s aims and scope. The work presented is original, however  the paper needs a number of clarifications/additions.

In greater detail the following should be corrected/added:

Introduction 

Line 69-71 - How to understand that: simple method and the design of test specimen is simple?

What is the aim of this study?

Materials and experimental program

Cement strength class not specified. What fine and coarse aggregates were used (specify them: particle size, bulk density and particle density). The same note on particle size for BFS and SF. What quantity was added SP and AE in concrete mixture?

What the authors had in mind when saying that - Line 119 - to achieve the enought workability? What is slump result of the concrete mix?

Not correct Table 1 name - Mix property of HSC because authors shows the composition of the concrete. Correct units to concrete Kg/m3 (should be -  kg/m3).

Not correct Table 2 name - Strength properties of concrete because authors shows the mechanicals properties of the concrete.

The authors did not specify in Table 2 after how many of curing days was determined properties of concrete.

I recommend increasing the Figure 6 size.

Author Response

(The authors gave the same response as above.)

Reviewer 5 Report

Comments

This paper studied the performance assessment of the post-tensioned anchorage zone using high strength concrete considering confinement effect. The outcome is interesting for readers. However, there are several aspects that need to be improved. The reviewer can only recommend for publication if the author satisfactorily address the following comments in the revised version.

  1. In Table 2, the splitting tensile strength of the concrete seems low compare with its compressive strength. What’s the reason behind that? Generally it is around 10% of compressive strength.
  2. Can the author provide a test setup photo for low velocity impact?
  3. The failure mechanism of the specimen should be discussed more clearly.
  4. The novelty of the study should be highlighted at the end of introduction section. How this study is different from the published study in literature?
  5. How the outcome of this study will benefit researchers and end users? This need to be highlighted in introduction or end of conclusion.
  6. The background study on the confinement effect is insufficient. Recently, the confinement effect was studied in reinforced concrete column [Ref: Hollow concrete columns: Review of structural behavior and new designs using GFRP reinforcement] and railway sleeper application [Ref: Static behaviour of glass fibre reinforced novel composite sleepers for mainline railway track]. Suggest to include them in introduction section with proper citations to improve the background study.

I would be happy to see the revised version to understand how these comments are being addressed.

Author Response

(The authors gave the same response as above.)

Round 2

Reviewer 1 Report

The manuscript readability has been improved, so I recommend to accept the article for publication in Materials. However, many stylistic and grammatical errors still remain in the article and need to be corrected before publication.